# Fiber Consumption Mediates Differences in Several Gut Microbes in a Subpopulation of Young Mexican Adults

**DOI:** 10.3390/nu14061214

**Published:** 2022-03-13

**Authors:** Avilene Rodríguez-Lara, Julio Plaza-Díaz, Patricia López-Uriarte, Alejandra Vázquez-Aguilar, Zyanya Reyes-Castillo, Ana I. Álvarez-Mercado

**Affiliations:** 1Institute of Nutrition and Food Technology, Biomedical Research Center, University of Granada, 18016 Armilla, Spain; avilenerl@correo.ugr.es (A.R.-L.); alejandravqz@correo.ugr.es (A.V.-A.); 2Department of Biochemistry and Molecular Biology II, School of Pharmacy, University of Granada, 18071 Granada, Spain; 3Children’s Hospital of Eastern Ontario Research Institute, Ottawa, ON K1H 8L1, Canada; 4Departamento de Ciencias Exactas y Metodologías del Centro Universitario del Sur, Universidad de Guadalajara, Ciudad Guzmán 49000, Mexico; patricia.lopezu@cusur.udg.mx; 5Instituto de Investigaciones en Comportamiento Alimentario y Nutrición, Universidad de Guadalajara, Ciudad Guzmán 49000, Mexico; zyanya.reyes@cusur.udg.mx; 6Instituto de Investigación Biosanitaria IBS.GRANADA, Complejo Hospitalario Universitario de Granada, 18014 Granada, Spain

**Keywords:** fiber consumption, gut microbes, body weight, young adults

## Abstract

Diet is a determinant for bodyweight and gut microbiota composition. Changes in dietary patterns are useful for the prevention and management of overweight and obesity. We aim to evaluate diet behavior and its potential association with selected gut bacteria and body weight among Mexican young adults. Mexican college students aged between 18 and 25 (normal-weight, overweight, and obese) were recruited. Anthropometric variables were recorded. A validated food frequency questionnaire was applied to all the participants. The percentages of macronutrients, fiber, and energy were calculated, and fecal samples were analyzed by real-time-qPCR to quantify selected gut bacteria. All the participants showed an unbalanced dietary pattern. However, the consumption of fruits, non-fat cereals, and oils and fats without protein were higher in the normal-weight individuals. In the overweight/obese participants, fiber intake did not correlate with the microbial variables, while Kcal from protein and *Clostridium leptum* correlated positively with *Lactobacillus*. Similarly, *Clostridium coccoides-Eubacterium rectale* correlated with *Akkermansia muciniphila.* In the normal-weight participants, *Clostridium leptum* and *Lactobacillus* correlated positively with *Clostridium coccoides-Eubacterium rectale* and *Bifidobacterium*, respectively, and *Bacteroidetes* negatively with *Akkermansia muciniphila*. In conclusion, a higher fiber intake had a positive impact on body weight and bacterial gut composition in this Mexican population of college students.

## 1. Introduction

Overweight and obesity are considered major public health concerns as they are key risk factors for the development of non-communicable diseases (which together cause over 60% of total mortality globally) [1].

The main cause of obesity and weight gain is a positive energy balance consequence of an increased energy intake and a decreased energy output associated with a loss of physical activity [2]. Thus, diet is one of the major determinants for body weight gain as well as a key tool in the prevention, management, and treatment of overweight and obesity [3]. In this context, the insufficient intake of whole grains, fruits, and vegetables, but abundant intake of discretionary foods, such as sugar-sweetened beverages, is ubiquitous, particularly in populations under 30 years old [4]. Accordingly, in recent decades, young adults aged 18–24 are gaining weight faster than their former generations. Indeed, weight gain is prominent among those aged 18–35 in most developed countries [4].

In the students’ population, this phenomenon might be related to the great change in all aspects that university life brings about.

Transition to university in young adults involves a greater autonomy concerning food choices, low food budget, and exposure to new social groups than usual [5,6]. Indeed, the transition from living at home with parents to autonomous university life is often associated with changes such as an increase in alcohol and sugar intake while at the same time a decrease in the consumption of fruits and vegetables [7,8,9].

On the other hand, in the last two decades the gastrointestinal (GI) microbiota (known as the community of microorganisms that subsists within the GI ecosystem) has emerged as a contributor to obesity and obesity-associated diseases [10].

A balanced bacterial composition is important for maintaining intestinal immunity and homeostasis. In healthy individuals, the role of the GI microbiota is to maintain a dynamic balance with the host, playing both local and remote functions in physiological processes such as inflammation and modulation of the immune response [11]. In contrast, an altered GI microbiota profile, referred to as dysbiosis, is found in obesity [12,13,14] and other metabolic diseases (e.g., type II diabetes mellitus and cardiovascular disease [15,16]).

Obese individuals showed different GI microbiota profiles than leans, *Bacteroidetes*, *Firmicutes*, and *Actinobacteria* being the most abundant phyla [16]. A greater *Firmicutes*/*Bacteroidetes* ratio, *Firmicutes*, *Fusobacteria*, *Proteobacteria*, *Mollicutes*, *Lactobacillus* (strains *L. reuteri*, *L. plantarum*, and *L. paracasei*, among others), and less *Verrucomicrobia* (*Akkermansia muciniphila*), *Faecalibacterium prausnitzii*, *Bacteroidetes*, and *Methanobrevibacter smithii* relative abundances have been found in obesity. Moreover, some bacteria present a positive correlation and others a negative correlation with obesity [17].

Having all this in mind, the increased interest in targeting the GI microbiome for the treatment and prevention of obesity is understandable. In this regard, one of the major influences on the microbial signatures of individuals is diet [18], and one way to alter the microbiome is through an increment in dietary fiber intake [19]. According to the United States Food and Drug Administration (FDA), dietary fiber is a carbohydrate component of an edible plant that is resistant to digestion and absorption. Thus, dietary fiber cannot be digested by the host but can be fermented by gut bacteria in the distal intestine, resulting in the production of short-chain fatty acids (SCFAs), which are known to benefit energy homeostasis and metabolism [19].

Nevertheless, to this date, to what extent an intervention with fiber may impact the human GI microbiota and therefore the metabolic regulation is not completely described [20], and it is mandatory, before developing an intervention, to explore as many contributory factors as possible for the entire population in general and for the university young adults in this particular subpopulation. Hence, it is important to analyze the behavior of food consumption together with the other social, biological, and psychological factors with a potential impact on the development of obesity. In consequence, the purpose of this study was to explore diet behavior among Mexican college students and their relationship with the proliferation of various bacterial rows based on body weight. We addressed this issue from nutritional and biological perspectives.

## 2. Materials and Methods

### 2.1. Study Design and Subjects

Subjects (females and males) aged between 18 and 25 years were selected in Ciudad Guzmán, a city located in Southern Mexico, from January to February 2014. Initially, 568 subjects were recruited in the first stage of the study, and then 50 subjects were divided according to anthropometric measures, such as body mass index (BMI) [21], into normal-weight and overweight/obese. The study was analytical, descriptive, and transversal. A schematic representation of this study is shown in Figure 1 This study was conducted in accordance with the Declaration of Helsinki and follows the rules of Law 14/2007 on Biomedical Research and the Organic Law 15/1999, RD 1720/2007 on the protection of personal data as well as international rules for research using samples from human beings. The participants’ accepted their inclusion in the study, signing the approved consent protocol. The confidentiality of the data obtained and any personal data used in this study has been kept and respected.

### 2.2. Variables and Data Collection

#### 2.2.1. Dietary Variables

A food frequency questionnaire (FFQ) validated for the Mexican population was applied [22]. A previous validation was carried out on 40 students from the University of Guadalajara in order to identify local foods and brands to be included in the questionnaire. The percentages of macronutrients (carbohydrates, lipids, proteins), fiber, ethanol, and food groups consumed were obtained after the FFQ analysis. With the information about all the major brands of fermented dairy foods that the participants consume, the colony-forming units (CFUs) variable was determined. A previous validation in 40 students from the University of Guadalajara was made to know the food groups in our population with a 24 h recall questionnaire. The food groups were: dairy products (whole milk, skimmed milk, yogurt), fruits (orange, banana, apple), vegetables (chard, spinach, lettuce), cereals with and without fats (potatoes, corn tortilla, rice, cookies, box bread), animal protein foods (egg, chicken, beef and pork), vegetable protein foods (chia, almonds, nuts), oils and fats with protein (butter, mayonnaise, lard), oils and fats without protein (olive oil, sunflower oil, canola oil), sugars (honey, table sugar, soft drink), and alcoholic beverages (beer, red wine, tequila) [23,24]. The fiber was calculated according to the ingested foods [25].

The participant made a detailed description of the dietary intake (ingredients, method of preparation, and brands); this information allowed the correct coding and weight assignment for each food item. The information obtained was structured as mealtimes (breakfast, mid-morning, lunch, mid-afternoon, dinner, and other moments), which helped us to calculate the distribution of energy and nutrients in the different moments of the day.

To analyze the individual FFQ, foods were grouped according to the United States Department of Agriculture and Human Services (USDA, 2010 [26]), as well as the Mexican System of Equivalents [27], calculating each of the servings (grams) per day for each participant. Likewise, for the CFU of fermented dairy foods, the brands registered by the Procuraduria Federal del Consumidor (PROFECO) [28] were taken as a reference. Total energy was calculated by multiplying the total in grams of each food group by the nutritional contribution of each macronutrient. It is worth mentioning that for the final analysis, a grouping by categorical BMI was performed, dividing the study groups into normal-weight and overweight/obese, to express the results in a general way by study group, presenting medians and percentiles for each of them.

#### 2.2.2. Anthropometric Variables

Body weight was measured in light clothes without shoes using Tanita BC-558 (Arlington Heights, IL, USA). Moreover, other variables were estimated with the aforementioned machine; they were Body fat percentage, Body water percentage, visceral fat percentage, muscle, basal metabolic rate, metabolic age, and bone mass. The participants were classified according to BMI, into overweight/obese (BMI > 25 kg/m^2^ and normal-weight (BMI = 18.5–24.9 kg/m^2^). In addition, a subgroup was created for the obese and overweight subjects.

### 2.3. Microbial Determination

#### 2.3.1. Stool Collection

Fecal samples were collected from each volunteer at the end of the evaluation using a sterile kit (including a plastic bottle with a screw cap, a tongue depressor, a glove, and a plastic bag with sealing). The fecal samples were placed inside the plastic bag and the bag was immediately sealed. Samples were kept at −20 °C and then transferred to −80 °C until analysis.

#### 2.3.2. DNA Extraction

The fecal samples were homogenized in a Stomacher-400 blender. DNA was extracted using a QIAamp DNA Stool Mini Kit (QIAGEN, Hilden, Germany) following the manufacturer’s instructions, with the exception that the samples were mixed with the lysis buffer and incubated at a temperature of 95 °C instead of 70 °C to ensure lysis of both the Gram-positive and the Gram-negative bacteria. Quantification was conducted using a NanoDrop ND-1000 spectrophotometer (Thermo Fisher Scientific, Kent, DE, USA) [29].

#### 2.3.3. Identification of the Intestinal Microbiota

The identification of the intestinal microbiota was carried out using specific and detailed primers (Real time-qPCR) with the Applied BioSystems StepOne platform (Waltham, MA, USA). The aforementioned primers, detailed in Table 1, were: *Firmicutes* (*Clostridium coccoides-Eubacterium rectale* and *Clostridium leptum*), *Bacteroidetes* (*Prevotella*, *Porphyromonas*, and *Bacteroides*), *Actinobacteria* (*Bifidobacterium* spp.), *Lactobacillus* (*Lactobacillus* spp.), and *Akkermansia muciniphila*. The PCR reaction was carried out under the following conditions: forward (F) and reverse (R) primers (0.2 µL of each at a concentration of 20 µM), 2 µL of Master Mix (Roche-Applied, Pleasanton, CA, USA), containing SYBR Green, MgCl_2_, Taq polymerase, and dNTP’s, DNA, and H_2_O to a final volume of 10 µL were placed individually according to the initial concentration of each sample to obtain a final concentration of 50 ng of DNA in each qPCR reaction. All samples were run in duplicate, and in cases where the duplicate gave a deviation greater than 2 cycles, the reaction was repeated. All target species and PCR program cycles (temperatures/times) are included in Table 1. Primers were synthesized by Invitrogen Life Technologies (Waltham, MA, USA), and were selected according to previously published articles with the idea of obtaining a validated assay for the detection of different bacterial species.

### 2.4. Statistical Analysis

Statistical tests were performed using IBM SPSS Statistics for Windows, Version 25.0 (IBM Corp., Armonk, NY, USA), and R version 3.6.1 (R Foundation for Statistical Computing, Vienna, Austria). Descriptive statistics were computed for each variable. All results are expressed as the mean ± standard deviation unless otherwise indicated. All P values were two-tailed, and statistical significance was considered at the 5% level (*p* < 0.05). To calculate the food consumption of the participants, a data dump was performed in the SPSS program, giving a coding for each response where: (never or almost never = 0, 1–3 times per month = 0.05, 1 time per week = 0.14, 2–4 times per week = 0.42, 5–6 times per week = 0. 78, 1 time per day = 1, 2–4 times per day = 3, 5–6 times per day = 5.5, and more than 6 times per day = 7, expressed in grams); this value was obtained by dividing the average number of times as appropriate (day, week, or month) by 1, 7, and 30 (e.g., 1–3 times per week = 2/30 = 0.06) to obtain the grams consumed per day.

Once each FFQ was coded, the grams consumed per day for each participant were calculated by multiplying the coding of each response by the amount in grams of each of the foods specified in the questionnaire. Subsequently, the foods were grouped according to the USDA to calculate the number of servings consumed per food group by each participant [26].

For the analysis of the fermented dairy products, each product was coded with the CFU reported by the brands asked; however, they were expressed in grams; so, the conversion of CFU contained in the grams of each product was performed and multiplied by the grams asked in the survey. The calculation of fiber was made taking into account the foods that contained more than 3 g of fiber of the FFQ applied, taking the Sistema Mexicano de Equivalentes [27] as a reference, and multiplied by the grams consumed per day by each participant.

Associations between dietary and microbial variables were tested using the Spearman correlation; these associations have shown significant variables, highlighted in red (negatively correlated) or blue (positively correlated), and the findings were corrected for multiple testing using the Benjamini–Hochberg procedure [34], using the corrplot function from the R studio [35]. Finally, differences between microbial variables were calculated using the U-Mann–Whitney test.

## 3. Results

### Participants

In both groups, the sociodemographic and anthropometric data were analyzed (Table 2). No changes were observed in age and gender distribution. The anthropometric data showed differences between normal-weight and overweight/obese in all variables with the exception of height (Table 2).

According to the food groups, changes were observed in fruits, cereals without fat, and oils and fats without protein. All these variables were higher in the normal-weight participants compared with overweight/obese (Table 3).

In the case of total energy, no differences were found between the groups, nor Kcal of different macromolecules and daily consumption calculated in grams. When the data were calculated according to recommendations of the USDA [21,29], the percentage of daily consumption was different for carbohydrates, lipids, and proteins (Table 4).

Microbial changes were observed in *Clostridium coccoides*-*Eubacterium rectale* in the comparison between normal-weight and overweight/obese participants (Figure 2).

The correlations between the microbial and dietary variables have shown that fiber consumption, CFU, Kcal from protein, Kcal from carbohydrates, and total energy were correlated negatively with the *Bacteroidetes* level in the normal-weight group. In the same group, some microbial variables were associated; *Clostridium leptum* and *Lactobacillus* were correlated positively with *Clostridium coccoides*-*Eubacterium rectale* and *Bifidobacterium*, respectively. Finally, *Bacteroidetes* was correlated negatively with *Akkermansia muciniphila*.

In the case of overweight/obese participants, fiber and CFU do not correlate with the microbial variables; only Kcal from protein was correlated positively with *Lactobacillus*. Here, *Bifidobacterium* was positively correlated with *Clostridium leptum* and *Lactobacillus* and, in the same way, *Clostridium coccoides*-*Eubacterium rectale* and *Akkermansia muciniphila* (Figure 3).

## 4. Discussion

Diet is one of the major determinants for body weight gain as well as a key tool in the prevention, management, and treatment of overweight and obesity. Insufficient intake of whole grains, fruits, and vegetables, but abundant intake of discretionary foods, such as sugar-sweetened beverages, are globally extended, particularly in young adults and college students [37,38]. Furthermore, a balanced microbial composition is important for maintaining intestinal immunity and homeostasis [39], while an altered gut intestinal microbiota is found in obesity and other metabolic diseases [40]. In addition, it is well known that one of the major influences on the microbial signatures of individuals is diet [41], and one way to alter the microbiome is through an increment in dietary fiber intake [42].

Bearing in mind that which is mentioned above, in the present study we evaluated diet behavior among Mexican college students and their relationship with the proliferation of various bacterial rows based on body weight, addressing this issue from nutritional and biological perspectives.

Our results show that both the normal-weight and overweight/obese groups are above the recommended dietary intakes (RDI) according to the percentage of adequacy in the consumption of alcoholic beverages, sugar, dairy products, grains and cereals, vegetables, and fruits. Besides, the consumption of fiber, oils, fats, and protein foods is below the RDI [43]. Neither of the total energy differences was found between the groups, which is in accordance with the results reported by Koo et al. (2019) [44]. These results indicate that both groups have an unhealthy diet and are at risk of presenting metabolic alterations, reflecting that adequate or healthy nutrition is not a priority at this stage of their lives. In accordance, college students tend to choose foods by cost and the ease of obtaining and consuming them; these are predominantly industrialized foods made by unhealthy preparations (e.g., fried), high in carbohydrates, fat, and energy and with low nutritional quality. Additionally, the influence of economics on food selection was evidenced by the fact that the consumption of proteins, which are generally more expensive, was markedly below the RDI. By contrast, more economical foods, such as vegetables, fruits, and sugars, were consumed to a greater extent. Moreover, and as reported by Sogari [45] et al., these strata of the population tend to migrate to have access to university, which is usually the first stage of life where they manage their food by themselves, which, together with the limited time available, can affect food selection and the establishment of adequate dietary patterns [45].

Nevertheless, significant differences were found between the groups (normal-weight and obese/overweight) in the consumption of fruits, non-fat cereals, and oils and fats without protein, reflecting the dietary contrasts in the sample studied. Consequently, this may influence to a greater or lesser extent the presence of obesity and metabolic disorders, which can also be reflected in the fact that the anthropometric data show differences between normal-weight and overweight/obese in all the variables with the exception of height.

On the other hand, the most common organisms in human gut microbiota are members of the Gram-positive *Firmicutes* and the Gram-negative *Bacteroidetes* phyla, with several other phyla, including the *Actinobacteria*, *Fusobacteria*, and *Verrucomicrobia*, that are present at subdominant levels [46]. Studies on the human intestinal microbiota have shown that obesity is associated with a reduction in Gram-negative bacteria, specifically members of the *Bacteroidetes* phyla [47]. Additionally, *Lactobacillus* can reduce body weight and alleviate fat accumulation in mice fed with a high-fat diet [48]. On the other hand, *Firmicutes* and *Actinobacteria* are the main responders to dietary fiber [49]. Body composition has been associated with higher levels of *Akkermansia muciniphila*, which may mediate the effects of dietary fiber [50].

Consequently, we selected the phyla *Bacteroidetes (Prevotella*, *Porphyromonas*, *and Bacteroides)*, *Firmicutes (Clostridium coccoides-Eubacterium rectale and Clostridium leptum)*, *Actinobacteria (Bifidobacterium* spp.), *Lactobacillus* spp., and *Akkermansia muciniphila* to compare the composition and expression of the intestinal microbiota of overweight/obese vs. normal-weight students, as well as the changes potentially associated with fiber intake, using bacterial probes that could identify several species (Table 1).

Interestingly, the phyla belonging to *Firmicutes* were found to be mostly expressed in overweight and obese individuals, with mainly *Clostridium coccoides-Eubacterium rectale* revealing a higher level in the composition of the intestinal microbiota of the overweight and obese individuals compared to the normal-weight subjects. Similar results have been found in a randomized clinical trial where *Blautia*, *Romboutsia*, *Ruminococcus2*, *Clostridium sensu stricto*, and *Dorea* were positively correlated with indicators of bodyweight [51].

In addition, the correlations between the microbial and dietary variables have shown that fiber consumption primarily in the form of non-fat cereal, fermented dairy foods, Kcal from protein, Kcal from carbohydrates, and total energy was correlated negatively with the *Bacteroidetes* level in the normal-weight group. In this respect, Wu and colleagues reported that high fecal *Bacteroides* abundance was positively associated with a protein- and animal-fat-rich diet and negatively with fiber [52], while other studies indicate that a fiber diet increases *Bacteroides* [53]. Moreover, in a large study conducted by Menni et al. (2017) fiber intake was positively correlated with measures of microbiome diversity; the conclusion was that gut microbiome diversity and high-fiber intake are related to lower long-term weight gain [54]. Other authors reported that although they found differences in the gut microbiome in obese individuals, fiber and fat/saturated fat diets were not key for central obesity [44].

Discrepancies could be attributed to the extent of the diet (short-term versus long-term). Given these controversial results, we agree with the reflection of Johnson and colleagues that reconciling the long-term and population-level patterns with the short-term observations requires a better understanding of every diet influence on digestive features such as changes in bile, pH, or substrate availability [47]. Moreover, the inconsistencies and contradictions between findings might also be explained because the influence of confounding factors, such as the composition of the diet, the energy content of the diet, the use of antibiotics, food availability, geographical areas, or age, are all factors that affect the gut microbial composition.

In the overweight/obese participants, fiber intake, and CFU do not correlate with the microbial variables, while Kcal from protein and *Clostridium leptum* was correlated positively with *Lactobacillus*. In the same way, *Clostridium coccoides*-*Eubacterium rectale* correlated with *Akkermansia muciniphila.* In the normal-weight participants, *Clostridium leptum* and *Lactobacillus* were correlated positively with *Clostridium coccoides*-*Eubacterium rectale* and *Bifidobacterium*, respectively, and *Bacteroidetes* was correlated negatively with *Akkermansia muciniphila*. In patients with type 2 diabetes, *A. muciniphila* was negatively associated with hemoglobin A1c [55]; its relative abundance is negatively associated with body mass [32,56,57,58].

Altogether, these results show a relationship between food consumption and the composition of the intestinal microbiota. This supports the fact that the gut microbiota composition responds to dietary patterns determined, among others, by the competition of the substrates that the bacterial species obtained from the diet.

In sum, the analysis of dietary intake from a biological and nutritional perspective, including a detailed study of energy, macronutrients, fiber, and CFU intake, and their relationship with gut microbiota and body weight, is the main contribution of this work. However, one of the main limitations of this study is the cross-sectional design which limited us to establishing an association between the variables at a specific time, but we were not able to establish cause–effect relationships between diet and the composition of the intestinal microbiota. It is also worthy to note that this study did not consider factors such as physical activity, social status, and the extent of obesity in time. Finally, the bacterial probes that were used in the study were selected in the literature, and they could be named as the *Lactobacillus* spp. or the *Clostridium leptum* group; however, these groups could include different species with several identification ratios.

We are aware that the bacterial population of the intestine has been performed by qPCR and not by 16S rRNA gene sequencing, the most-used sequence-based bacterial analysis for decades, which would give a broader view to this study. However, we believe that the selection of the analyzed bacteria has been sufficiently justified and our results sufficiently conclusive.

Another very interesting aspect that remains to be evaluated is the difference between the subgroups of obese and overweight. However, the number of subjects available was very limited, which did not allow us to carry out these analyses.

Nevertheless, although other studies have already reported alterations in the intestinal microbiota related to body weight, this study not only corroborates those results but extends them to a well-established population of young university students. Our study was conducted in a very homogeneous population in a well-limited geographical area, which limited the bias due to geographical origin–location.

Importantly, the participants are also in an important lifetime period, the transition from youth to maturity when newly acquired habits concerning diet and their effects may be crucial for the rest of their lives.

Consequently, the more knowledge we have the better to design appropriate dietary interventions in a specific context. Such strategies should not be limited only to variables related to energy and macronutrient intake but should be redirected to the integral analysis of nutrition in a multidisciplinary manner, considering aspects such as genetics, environment, and lifestyles, among others.

To conclude our study, we revealed an unbalanced dietary pattern in the normal-weight and overweight/obese subjects in a subpopulation of Mexican young adults. The phyla belonging to *Firmicutes* were found to be mostly expressed in overweight and obese individuals, indicating an alteration in the composition of the intestinal microbiota of these individuals compared to the normal-weight subjects. Increased fiber intake in the participants influences to a lesser extent the presence of obesity and overweight and a bacterial gut composition more associated with health.

## Figures and Tables

**Figure 1 nutrients-14-01214-f001:**
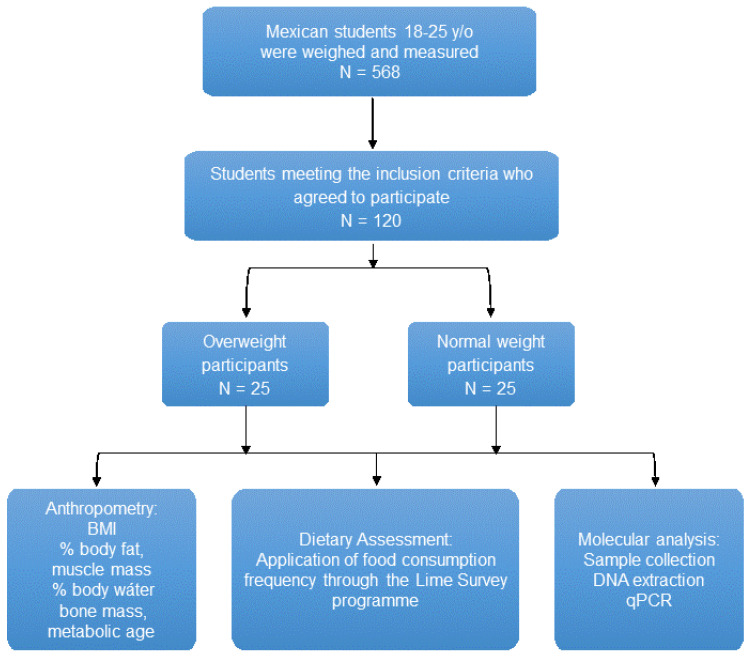
Schematic diagram of the present study. Abbreviations: BMI, body mass index; DNA, deoxyribonucleic acid; qPCR, quantitative polymerase chain reaction; y/o, years old; %, percentage.

**Figure 2 nutrients-14-01214-f002:**
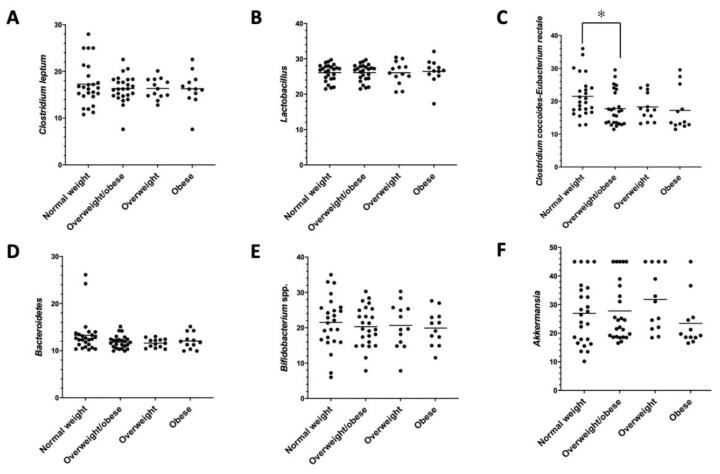
Microbial differences by qPCR. (**A**) *Clostridium leptum*, (**B**) *Lactobacillus* spp., (**C**) *Clostridium coccoides-Eubacterium rectale*, (**D**) *Bacteroidetes*, (**E**) *Bifidobacterium* spp., and (**F**) *Akkermansia muciniphila*. * *p* < 0.05, normal-weight vs. overweight/obese subjects.

**Figure 3 nutrients-14-01214-f003:**
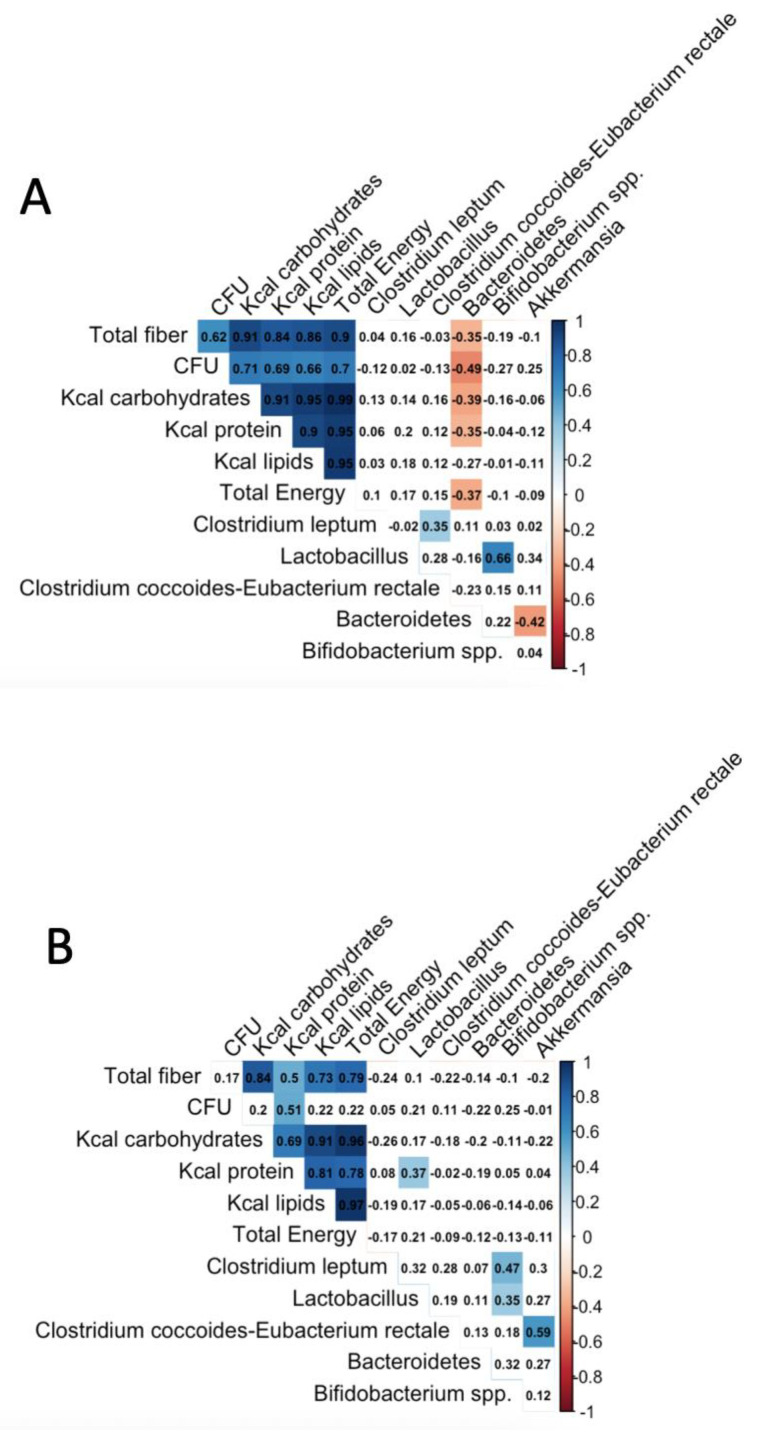
Spearman correlations between nutritional and microbial variables. (**A**) Normal-weight subjects, (**B**) overweight/obese subjects. Associations between dietary and microbial variables were tested using Spearman correlation; these associations have shown significant variables highlighted in red (negatively correlated) or blue (positively correlated), and findings were corrected for multiple testing using the Benjamini–Hochberg procedure [34], using the corrplot function from the R studio [35]. Abbreviations: BMI, body mass index; qPCR, quantitative polymerase chain reaction.

**Table 1 nutrients-14-01214-t001:** Sequences of primers for real-time qPCR.

PCR Assay	Sequence	Target Species	Size (bp)	Cycles and Tm (°C)	Reference
*Bacteroidetes (Bacteroides*, *Prevotella and Porphyromonas)*	F. 5′-GGTGTCGGCTTAAGTGCCAT-3′, R. 5′-CGGA(C/T)GTAAGGGCCGTGC-3′	*Bacteroides fragilis*, *B. stercoris*, *B. vulgatus*, *B. eggerthii*, *B. acidofaciens*, *B. caccae*, *B. ovatus*, *B. uniformis*, *B. thetaiotaomicron*, *B. distasonis*, *B. merdae*, *B. forsythus*, *Prevotella tannerae*, *P. bryantii*, *P. ruminicola*, *P. heparinolytica*, *P. zoogleoformans*, *P. brevis*, *P. loescheii*, *P. buccae*, *P. oralis*, *P. enoeca*, *P. melaninogenica*, *P. veroralis*, *P. intermedia P. albensis*, *P. nigrescens*, *P. corporis*, *P. disiens*, *P. bivia*, *P. pallens*, *P. denticola*, *Porphyromonas canoris*, *P. gingivalis*, *P. asaccharolytica*, *P. levii*, *P. cangingivalis*, *P. endodontalis*, *P. macacae*, *P. circumdentaria*, *P. catoniae*	140	Polymerase activation at 95 °C for 10 min, and 45 cycles of denaturation (95 °C/10 s), then annealing (68 °C/8 s), and extension (72 °C/6 s); 68 °C	Rinttilä, (2004) [30]
*Actinobacteria (Bifidobacterium* spp.)	F. 5′-TCGCGTC(C/T)GGTGTGAAAG-3′, R. 5′-CCACATCCAGC(A/G)TCCAC-3′	*Bifidobacterium longum*, *B. minimum*, *B. angulatum*, *B. catenulatum*, *B. pseudocatenulatum*, *B. dentium*, *B. ruminantium*, *B. thermophilum*, *B. subtile*, *B. bifidum*, *B. boum*, *B. lactis*, *B. animalis*, *B. choerinum*, *B. gallicum*, *B. pseudolongum subsp. globosum*, *B. pseudolongum subsp. pseudolongum*, *B. magnum*, *B. infantis*, *B. indicum*, *B. gallinarum*, *B. pullorum*, *B. saeculare*, *B. suis*	243	Polymerase activation at 95 °C for 10 min, and 45 cycles of denaturation (95 °C/10 s), then annealing (58 °C/8 s), and extension (72 °C/10 s); 58 °C	Rinttilä, (2004) [30]
*Firmicutes (Clostridium coccoides-Eubacterium rectale)*	F. 5′-CGGTACCTGACTAAGAAGC-3′, R. 5′-AGTTT(C/T)ATTCTTGCGAACG-3′	*Clostridium coccoides*, *C. proteoclasticum*, *C. aminophilum*, *C. symbiosum*, *C. sphenoides*, *C. celerecrescens*, *C. aerotolerans*, *C. xylanolyticum*, *C. clostridiiforme*, *C. fusiformis*, *C. nexile*, *C. oroticum*, *C. populeti*, *C. aminovalericum*, *C. indolis*, *C. herbivorans*, *C. polysaccharolyticum*, *Eubacterium xylanophilum*, *E. ruminantium*, *E. saburreum*, *E. fissicatena*, *E. hadrum*, *E. rectale*, *E. ramulus*, *E. contortum*, *E. eligens*, *E. hallii*, *E. formicigenerans*, *E. cellulosolvens*, *Ruminococcus productus*, *R. obeum*, *R. schinkii*, *R. hydrogenotrophicus*, *R. hansenii*, *R. torques*, *R. lactaris*, *R. gnavus*, *Butyrivibrio fibrisolvens*, *B. crossotus*, *B. fibrisolvens*, *Desulfotomaculum guttoideum*, *Roseburia cecicola*, *Pseudobutyrivibrio ruminis*, *Lachnospira multipara*, *L. pectinoschiza*, *Acetitomaculum ruminis*, *Catonella morbi*	429	Polymerase activation at 95 °C for 10 min, and 45 cycles of denaturation (95 °C/10 s), then annealing (58 °C/8 s), and extension (72 °C/10 s); 58 °C	Rinttilä, (2004) [30]
*Firmicutes (Clostridium leptum)*	F. 5′-GCA CAA GCA GTG GAGT-3′, R. 5′-CTT CCT CCG TTT TGT CAA-3′	*Clostridium leptum*, *C. viride*, *Eubacterium siraeum*, *Ruminococcus bromii*, *R. callidus*, *R. albus*	239	Polymerase activation at 95 °C for 10 min, and 45 cycles of denaturation (95 °C/10 s), then annealing (58 °C/8 s), and extension (72 °C/14 s); 50 °C	Matsuki, (2004) [31]
*Akkermansia*	F.5′ CAGCACGTGAAGGTGGGAC-3′, R. 5′-CCTTGCGGTTG GCTTCAGAT-3′	*Akkermansia muciniphila*		Polymerase activation at 95 °C for 10 min, and 45 cycles of denaturation (95 °C/10 s), then annealing (58 °C/8 s), and extension (72 °C/30 s); 62 °C	Dao, (2016) [32]
*Lactobacillus*	F.5′-AGCAGTAGGGAATCTTCCA-3′, R. 5′-CACCGCTACACATGGAG-3′	*Lactobacillus acidophilus*, *L. amylovorus*, *L. delbrueckii* subsp. *bulgaricus*, *L. delbrueckii* subsp. *delbrueckii*, *L. delbrueckii* subsp. *lactis*, *L. amylolyticus*, *L. acetotolerans*, *L. crispatus*, *L. amylophilus*, *L. johnsonii*, *L. gasseri*, *L. fermentum*, *L. pontis*, *L. reuteri*, *L. mucosae*, *L. vaginalis*, *L. panis*, *L. oris*, *L. pentosus*, *L. plantarum*, *L. collinoides*, *L. alimentarius*, *L. farciminis*, *L. brevis*, *L. buchneri*, *L. kefiri*, *L. fructivorans*, *L. mali*, *L. animalis*, *L. murinus*, *L. ruminis*, *L. agilis*, *L. salivarius* subsp. *salicinius*, *L. aviarius* subsp. *aviarius*, *L. sharpeae*, *L. manihotivorans*, *L. rhamnosus*, *L. casei* subsp. *casei*, *L. zeae*, *L. paracasei* subsp. *paracasei*, *L. paracasei* subsp. *tolerans*, *L. coryniformis* subsp. *coryniformis*, *L. bifermentans*, *L. perolens*, *L. sakei* subsp. *sakei*, *L. casei* subsp. *fusiformis*, *Pediococcus pentosaceus*, *P. parvulus*, *P. acidilactici*, *P. dextrinicus*, *Weissella halotolerans*, *W. confusus*, *W. Paramesenteroides*, *W. hellenica*, *W. viridescens*, *W. kandleri*, *W. minor*, *Leuconostoc lactis*		Amplification program was 92 °C for 2 min, followed by 40 cycles of 95 °C for 30 s, 30 s at the appropriate annealing temperature, and 72 °C for 30 s; 56 °C	Walter et al., (2001) [33]

Abbreviations: bp, base pair; F, forward; min, minutes; qPCR, quantitative polymerase chain reaction; R, reverse; s, seconds; Tm, primer melting temperature.

**Table 2 nutrients-14-01214-t002:** Sociodemographic and anthropometric characteristics of the study subjects.

Characteristics	Normal-Weight *n* = 25	Overweight/Obese *n* = 25	Overweight *n* = 13	Obese *n* = 12	*p*-Value
**Sociodemographic Data**					
Age, years	20.5 ± 1.7	20.7 ± 1.7	21 ± 1.6	21 ± 1.8	NS
Sex, F/M	15/10	12/13	6/7	6/6	NS
**Anthropometric Data**					
Weight (Kg)	59.0 ± 8.0	85.0 ± 12.0	80.0 ± 9.0	89.0 ± 14.0	0.0001
Height (cm)	165.0 ± 9.0	169.0 ± 9.0	170.0 ± 9.0	167.0 ± 10.0	NS
BMI, kg/m^2^	21.9 ± 1.9	29.6 ± 3.7	27.7 ± 2.1	31.8 ± 4.0	0.0001
Body fat, %	23.3 ± 6.7	32.5 ± 7.1	31.5 ± 7.3	33.6 ± 7.0	0.0001
Body water, %	56.9 ± 4.8	51.7 ± 4.8	51.3 ± 4.5	52.1 ± 5.3	0.003
Visceral fat, %	1.6 ± 0.8	5.4 ± 2.7	4.8 ± 1.9	6.0 ± 3.4	0.0001
Muscle	43.5 ± 8.3	53.8 ± 11.1	52.3 ± 11.5	55.5 ± 10.9	0.0007
Basal metabolic rate	1432.0 ± 226.0	1704.0 ± 330.0	1676.0 ± 297.0	1735.0 ± 374.0	0.002
Metabolic age	17.4 ± 6.3	44.8 ± 13.7	42.9 ± 10.2	46.7 ± 16.9	0.0001
Bone mass	2.36 ± 0.4	3.1 ± 1.1	3.2 ± 1.5	3.1 ± 0.5	0.0001

Variables are expressed as mean ± standard deviation. Statistical differences were calculated with U-Mann–Whitney test (normal-weight vs. overweight/obese). Abbreviations: BMI, body mass index; F, female; M, male; NS, not significant.

**Table 3 nutrients-14-01214-t003:** Consumption of food portions of the study groups according to the Food Guide recommended intake for Americans 2010 [26,36].

Portions/Day	Normal-Weight *n* = 25	Overweight/Obese *n* = 25	*p*-Value
Dairy products	3.8 (0.4–13.8)	3.8 (1.0–10.9)	0.915
Fruits	7.5 (3.0–14.8)	5.2 (1.5–10.5)	0.043
Vegetables	9.5 (3.5–15.3)	6.3 (2.8–13.0)	0.066
Cereal with fat	4.9 (2.6–8.7)	3.9 (0.8–9.8)	0.132
Cereal without fat	6.0 (3.1–11.2)	5.0 (2.3–6.4)	0.005
Animal protein foods	2.2 (1.4–4.9)	2.1 (0.5–4.9)	0.455
Vegetable protein foods	0.4 (0.1–1.4)	0.7 (0.1–1.9)	0.331
Oils and fats with protein	1.2 (0.3–5.6)	1.0 (0.0–7.7)	0.414
Oils and fats without protein	4.4 (1.6–8.9)	2.9 (1.0–5.8)	0.022
Sugars	6.4 (2.9–10.8)	6.5 (2.1–8.7)	0.472
Alcoholic beverages	1.4 (0.0–2.5)	1.0 (0.0–2.5)	0.682
Fiber (g per day)	19.9 (9.5–37.2)	19.0 (7.4–35.8)	0.366
Fermented dairy foods (CFU × 10^9^ per day)	18.3 (0.0–240.4)	29.6 (0–177.4)	0.661

Variables are expressed as several servings (median (p5–p95)), except for the fiber (g per day). Statistical differences were calculated with the U-Mann–Whitney test (normal-weight vs. overweight/obese). Abbreviations: CFU, Colony-forming unit.

**Table 4 nutrients-14-01214-t004:** Total energy and macronutrient intake in the study groups.

Variables	Normal Weight *n* = 25	Overweight/Obesity *n* = 25	*p*-Value
**Total energy**	3122 (1600–6843)	2825 (1002–4953)	0.232
kcal/day			
**Carbohydrates**	464 (240–1005)	394 (151–755)	0.063
g/day ^a^			
Kcal/day	1886 (969–4031)	1576 (604–3044)	0.066
% daily ^b^	60.4 (86–337)	55.7(44–210)	0.005
**Proteins**	95 (53–273)	101 (33–190)	0.763
g/day ^a^			
Kcal/day	382 (215–1093)	404 (133–761)	0.763
% daily ^b^	12.2 (76–349)	14.3 (41–221)	0.037
**Lipids**	86 (44–204)	80 (28–138)	0.377
g/day ^a^			
Kcal/day	774 (398–1841)	724 (252–1216)	0.377
% daily ^b^	24.7 (76–311)	25.6 (40–191)	0.017

Statistical differences were calculated with the U-Mann–Whitney test (normal-weight vs. overweight/obese). ^a^ Calculated based on the Sistema Mexicano de Equivalentes [27], ^b^ calculated according to the recommendations of the USDA [26,36].

## Data Availability

The datasets generated during and/or analyzed during the current study are available from the corresponding author upon reasonable request.

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
