# Peer review of "Fiber Consumption Mediates Differences in Several Gut Microbes in a Subpopulation of Young Mexican Adults"

_nutrients, 2022, doi:10.3390/nu14061214_

Round 1

Reviewer 1 Report

Lara et al. present a study in which they subjected 20 normal-weight and 20 lean individuals to anthropometrics (e.g. body fat determination) and gut microbiota composition after assessing their diet via a food frequency questionnaire. Gut microbiota composition was studied not via 16S rRNA shotgut sequencing, but via realtime-PCR and sonds targeting groups of human gut bacteria. The main outcome of this study was, that a slight change in gut microbiota composition was detected in overweight vs normal-weight people: A statistically significant lower abundance of a Firmicutes group in the overweight group. Connection of body weight to diet was minor, while a few correlations of microbiota composition to diet were suggested. 

First of all, I positively point out the discussed limitations and critical points of the study in the discussion part. This is good, especially given the contradicting results of a lot of previous studies on the same matter. While this study is just on of several ten thousands on a similar matter (diet, overweight and gut microbiota) and the study size is very small (2 x 20), the studied groups were relatively homogenous. However, I have several reservations regarding the methodological design, the result description (especially the microbiological part) and the conclusions of the study, which I will summarize in the major points below. 

1) I cannot find a reference to the food questionnaire and from the described method (L116ff), I cannot conclude on the timeframe you investigated here: was it indeed just a questionnaire for the last 24 hours, was the diet of the last days/weeks/months investigated? If only for 24 hours, could the "long-term" diet extrapolated and if so, how reliably? Please take all this into account when you present and discuss the data! This is especially important if the diet of the last 24 hours might have been different than in the period(s) before filling out the questionnaire, was something done to prevent biases?

2) Probably the most critical part of the manuscript is that relatively old primer probes were used to quantify the groups of gut bacteria. In principle, nothing speaks completely against using a realtime-PCR microbiota composition study with primer probes based on 16S-rRNA gene-specific stretches. However, these probes are often not very specific, especially when they were designed decades ago, as is the case with most primer pairs here! There is an online tool to check probe specificities, the authors did obviously not use this in their study. I checked the used probes via http://rdp.cme.msu.edu/probematch/search.jsp and it turned out that especially the two Firmicutes probes are not very specific. While the authors could use these, it is crucial to mention, that they're both detecting a huge group of different Firmicutes. The Clostridium leptum primers do not detect Clostridium leptum, but more other bacteria from genera as different as Eubacterium, Roseburia, Lachnospira, etc., etc. This should be noted everywhere in the manuscript. 

3) The details of the realtime PCR (primer melting curves, PCR program temperatures/times, etc.) are not given, but are crucial for understanding and following the study (some details can be included in a supplement, also the single realtime PCR data points should be given in the supplement!). Related to that, the values of the data points in figure 2 are cryptic: Are the values ranging from approximately 5 to 50 Ct values - so higher values mean indeed lower abundance of bacteria? What were the standards used? To me it seems that the realtime PCR results are not really trustable, because so few details are given. There is usually a standard curve with a given amount of bacterial cells corresponding to Ct values and I can't see that here. 

4) Figure 3 is missing, but also here it should be mentioned that a lot of different bacterial species are detected, the resolution of the bacterial probes is just too low to have meaningful discussions of what bacterial genera correlated with specific dietary compounds! Be more general, e.g. chose Firmicutes when you mention single bacterial species! 

5) Especially in the introduction, but also discussion, more works on gut microbiota and diet connected to obesity should be cited. I note that the authors included some good and recent reviews, but for example the lines from 60 to 72 cites only 4 references, but here, at a broader overview should be given (and discussed!)! Some reviews e.g. https://academic.oup.com/advances/article/10/suppl_1/S17/5307226

https://www.nature.com/articles/s41430-018-0306-8

https://www.tandfonline.com/doi/full/10.1080/19490976.2018.1465157

some original research (highly related to yours, so please discuss accordingly!): 

https://onlinelibrary.wiley.com/doi/full/10.1002/jgh3.12184

https://www.nature.com/articles/ijo201766

Minor:

The abstract and discussion are often not very specific, terms like "critical lifetime period" and "inadequate dietary patterns" should be avoided. 

Why are the p-values of different values in table 4 significant when the percentage (USDA recommendations) are given, but not in grams? Should a more strict test be chosen here? I am no statistician, but this seems odd. 

Author Response

COMMENTS FROM REVIEWER #1

Lara et al. present a study in which they subjected 20 normal-weight and 20 lean individuals to anthropometrics (e.g. body fat determination) and gut microbiota composition after assessing their diet via a food frequency questionnaire. Gut microbiota composition was studied not via 16S rRNA shotgun sequencing, but via realtime-PCR and sounds targeting groups of human gut bacteria. The main outcome of this study was, that a slight change in gut microbiota composition was detected in overweight vs normal-weight people: A statistically significant lower abundance of a Firmicutes group in the overweight group. Connection of body weight to diet was minor, while a few correlations of microbiota composition to diet were suggested.

First of all, I positively point out the discussed limitations and critical points of the study in the discussion part. This is good, especially given the contradicting results of a lot of previous studies on the same matter. While this study is just on of several ten thousands on a similar matter (diet, overweight and gut microbiota) and the study size is very small (2 x 20), the studied groups were relatively homogenous. However, I have several reservations regarding the methodological design, the result description (especially the microbiological part) and the conclusions of the study, which I will summarize in the major points below.

Response: Thank you very much for taking the time to review our work and for your constructive comments about our manuscript.

Comment #1

I cannot find a reference to the food questionnaire and from the described method (L116ff), I cannot conclude on the timeframe you investigated here: was it indeed just a questionnaire for the last 24 hours, was the diet of the last days/weeks/months investigated? If only for 24 hours, could the "long-term" diet extrapolated and if so, how reliably? Please take all this into account when you present and discuss the data! This is especially important if the diet of the last 24 hours might have been different than in the period(s) before filling out the questionnaire, was something done to prevent biases?

Response: We have added the reference to the food questionnaire following the reviewer's comment. We have included a sentence in the methodology section briefly explaining this observation. The manuscript now states (page 3, lines 112-115), “A food frequency questionnaire (FFQ) validated for the Mexican population was applied [22]. A previous validation was carried out on 40 students from the University of Guadalajara to identify local foods and brands to be included in the questionnaire”.

The 24-hour recall was only used in a small sample of students to learn about and identify the foods consumed in the region where the study was conducted. The tool for dietary assessment was the FFQ described above. Regarding the observation on the reliability of the tool used to assess food consumption, the frequency questionnaire used has a long-term time horizon (specifically one year backward). This avoids the bias of possible dietary change at the time of sample delivery concerning the completion of the questionnaire.

Comment #2

Probably the most critical part of the manuscript is that relatively old primer probes were used to quantify the groups of gut bacteria. In principle, nothing speaks completely against using a real-time-PCR microbiota composition study with primer probes based on 16S-rRNA gene-specific stretches. However, these probes are often not very specific, especially when they were designed decades ago, as is the case with most primer pairs here! There is an online tool to check probe specificities, the authors did obviously not use this in their study. I checked the used probes via http://rdp.cme.msu.edu/probematch/search.jsp and it turned out that especially the two Firmicutes probes are not very specific. While the authors could use these, it is crucial to mention, that they're both detecting a huge group of different Firmicutes. The Clostridium leptum primers do not detect Clostridium leptum, but more other bacteria from genera as different as Eubacterium, Roseburia, Lachnospira, etc., etc. This should be noted everywhere in the manuscript.

Response: Thanks to the reviewer for his/her comment about our manuscript. We understand the comment about the specificity of the different probes. After a literature search, these were the selected probes and they have shown in Table 1 with the new specifications.  In addition, putting all the species in Table 1 we would take the opportunity to show the readers the broad spectrum of the probes. Using the reviewer’s comment, the limitations add new information and now state (page 12, lines 371-374), “Finally, the bacterial probes that were used in the study were selected in the literature, and they could be named as Lactobacillus spp. or Clostridium leptum group, however, these groups could include different species with several identification ratios.”

Comment #3

The details of the real-time PCR (primer melting curves, PCR program temperatures/times, etc.) are not given, but are crucial for understanding and following the study (some details can be included in a supplement, also the single real-time PCR data points should be given in the supplement!). Related to that, the values of the data points in figure 2 are cryptic: Are the values ranging from approximately 5 to 50 Ct values - so higher values mean indeed lower abundance of bacteria? What were the standards used? To me it seems that the realtime PCR results are not really trustable, because so few details are given. There is usually a standard curve with a given amount of bacterial cells corresponding to Ct values and I can't see that here.

Response: Using the reviewer’s comment, the information about the process and the conditions is in Table 1 (Pages 5-6). The manuscript now states (pages 4-5, lines 170-177: “The PCR reaction was carried out under the following conditions, in a final volume of 10 µL: forward (F) and reverse (R) primers (0.2 µL of each at a concentration of 20 uM), 2 µL of Master Mix (Roche-Applied, USA); containing SYBR Green, MgCl2, Taq polymerase and dNTPʼs), DNA and H2O were placed individually according to the initial concentration of each sample with the aim of obtaining a final concentration of 50 ng of DNA in each qPCR reaction. All samples were run in duplicate and in cases where the duplicate gave a deviation greater than 2 cycles, the reaction was repeated.” All the probes were tested for 45 cycles with the exception of Lactobacillus spp. for 40 cycles. The results have shown in all probes a mean value of fewer than 30 cycles, an indication of the proper abundance of bacteria. Only in the case of Akkermansia muciniphila we have found samples without detection (Ct=45).

Comment #4

Figure 3 is missing, but also here it should be mentioned that a lot of different bacterial species are detected, the resolution of the bacterial probes is just too low to have meaningful discussions of what bacterial genera correlated with specific dietary compounds! Be more general, e.g. chose Firmicutes when you mention single bacterial species!

Response: We apologized for deleting Figure 3 during the submission process. Now, the manuscript shows Figure 3, and accordingly, to the reviewer’s comment, the discussion has also been modified.

Comment #5

5) Especially in the introduction, but also discussion, more works on gut microbiota and diet connected to obesity should be cited. I note that the authors included some good and recent reviews, but for example, the lines from 60 to 72 cite only 4 references, but here, a broader overview should be given (and discussed!)!

Some reviews e.g. https://academic.oup.com/advances/article/10/suppl_1/S17/5307226

https://www.nature.com/articles/s41430-018-0306-8

https://www.tandfonline.com/doi/full/10.1080/19490976.2018.1465157

some original research (highly related to yours, so please discuss accordingly!):

https://onlinelibrary.wiley.com/doi/full/10.1002/jgh3.12184

https://www.nature.com/articles/ijo201766

Response: We agree with the reviewer's commentary and certainly, the number of reviews dealing with the topic of diet, obesity, and microbiota is large and we may miss some of them. However, this paper is not a review of the topic in question where most of the research reported in the area should be considered. The works that we have selected, included, and discussed are those that we consider relevant and necessary to mention in order to put our work in context and justify our results. In any case, we have included the references that the reviewer suggested as you can see in the list of references. We have also included results reported by the reviewer in the discussion section as the reviewer advised us as follow:

(Page 11, lines 289-290), “Neither total energy differences were found between the groups which are in accordance with the results reported by Koo et al. (2019) [44].”

(Page 12, lines 337-341), “Also, in a large study conducted by Menni et al. (2017) fiber intake was positively correlated with measures of microbiome diversity concluding that gut microbiome diversity and high-fiber intake are related to lower long-term weight gain [54]. Other authors reported that although they found differences in the gut microbiome in obese individuals, fiber and fat/saturated fat diets were not key for central obesity [44].”

We have also added the following  references:

  1. Cuevas-Sierra, A.; Ramos-Lopez, O.; Riezu-Boj, J.I.; Milagro, F.I.; Martinez, J.A. Diet, Gut Microbiota, and Obesity: Links with Host Genetics and Epigenetics and Potential Applications. Adv Nutr 2019, 10, S17-S30, doi:10.1093/advances/nmy078

  1. Cornejo-Pareja, I.; Munoz-Garach, A.; Clemente-Postigo, M.; Tinahones, F.J. Importance of gut microbiota in obesity. Eur J Clin Nutr 2019, 72, 26-37, doi:10.1038/s41430-018-0306-8.
  2. Gomes, A.C.; Hoffmann, C.; Mota, J.F. The human gut microbiota: Metabolism and perspective in obesity. Gut Microbes 2018, 9, 308-325, doi:10.1080/19490976.2018.1465157.

Comment #6

The abstract and discussion are often not very specific, terms like "critical lifetime period" and "inadequate dietary patterns" should be avoided. 

Response: As the reviewer suggests we have changed “critical lifetime period” by “an important lifetime period like the transition from youth to maturity” (line 388-389).

“Inadequate dietary patterns” has been replaced by unbalanced (lines 26 and 396).

Comment #7

Why are the p-values of different values in table 4 significant when the percentage (USDA recommendations) are given, but not in grams? Should a more strict test be chosen here? I am no statistician, but this seems odd.

Response: Thank you to the reviewer for his/her comment about our manuscript. The differences are given because the data are calculated from different databases, in the case of significant values, they were calculated according to the recommendations of the USDA [1,2].

1.- Committee, U.S.D.G.A. Dietary guidelines for Americans, 2010; US Department of Health and Human Services, US Department of Agriculture: 2010.

2.- McGuire, S. US department of agriculture and US department of health and human services, dietary guidelines for Americans, 2010. Washington, DC: US government printing office, January 2011. Advances in nutrition 2011, 2, 293-294.

Reviewer 2 Report

The authors evaluated the diet habits as well as their potential association with gut bacteria and body weight in Mexican college students. Firstly, they developed a questionnaire approved by the ethical committee on the students habits, and thereafter they analyzed fecal samples  by real-time-qPCR to quantify selected gut bacteria . Obese students showed a different profile compared to normal weight ones. Normal weight students diet was correlated with consumption of fruits, non-fat cereals, and oils and fats without protein,while  in overweigh participants, fiber intake does not correlate with microbial variables ,while Kcal from protein and Clostridium leptum correlated positively with Lactobacillus. Yet,Clostridium coccoides-Eubacterium rectale correlated with Akkermansia muciniphila.

In contrast, in the normal weight,Clostridium leptum and Lactobacillus correlated positively with Clostridium coccoides-Eubacterium rectale and Bifidobacterium, and the Bacteroidetes correlated negatively with Akkermansia municiniphila.The authors concluded that a higher fiber intake diet seems to have a positive impact on the body weight and intestinal bacterial composition.

It is a well-written paper based both on the collection of anthropometric data and population habits enriched with molecular studies of identification of the intestinal microbiota.Statistical evaluation was also performed.Material and Methods is well organized and results are well investigated and discussed.

Author Response

Thank you very much for taking the time to review our work and for your kind and constructive comments about our manuscript.
